# Investigating the Human Intestinal DNA Virome and Predicting Disease-Associated Virus–Host Interactions in Severe Myalgic Encephalomyelitis/Chronic Fatigue Syndrome (ME/CFS)

**DOI:** 10.3390/ijms242417267

**Published:** 2023-12-08

**Authors:** Shen-Yuan Hsieh, George M. Savva, Andrea Telatin, Sumeet K. Tiwari, Mohammad A. Tariq, Fiona Newberry, Katharine A. Seton, Catherine Booth, Amolak S. Bansal, Thomas Wileman, Evelien M. Adriaenssens, Simon R. Carding

**Affiliations:** 1Food, Microbiome, and Health Research Programme, Quadram Institute Bioscience, Norwich Research Park, Norwich NR4 7UQ, UK; ernie.hsieh@quadram.ac.uk (S.-Y.H.); andrea.telatin@quadram.ac.uk (A.T.); sumeet.tiwari@quadram.ac.uk (S.K.T.); adnan.adnan.tariq@northumbria.ac.uk (M.A.T.); fiona.newberry@ntu.ac.uk (F.N.); katharine.seton@quadram.ac.uk (K.A.S.); tom.wileman@quadram.ac.uk (T.W.);; 2Core Science Resources, Quadram Institute Bioscience, Norwich NR4 7UQ, UK; george.savva@quadram.ac.uk (G.M.S.); catherine.booth@quadram.ac.uk (C.B.); 3Spire St. Anthony’s Hospital, Surrey SM3 9DW, UK; asbansal1000@yahoo.com; 4Norwich Medical School, University of East Anglia, Norwich NR4 7TJ, UK

**Keywords:** myalgic encephalomyelitis/chronic fatigue syndrome (ME/CFS), virus-like particle (VLP), virome, bacteriophage, viral operational taxonomic unit (vOTU), bacteriome

## Abstract

Understanding how the human virome, and which of its constituents, contributes to health or disease states is reliant on obtaining comprehensive virome profiles. By combining DNA viromes from isolated virus-like particles (VLPs) and whole metagenomes from the same faecal sample of a small cohort of healthy individuals and patients with severe myalgic encephalomyelitis/chronic fatigue syndrome (ME/CFS), we have obtained a more inclusive profile of the human intestinal DNA virome. Key features are the identification of a core virome comprising tailed phages of the class *Caudoviricetes*, and a greater diversity of DNA viruses including extracellular phages and integrated prophages. Using an in silico approach, we predicted interactions between members of the *Anaerotruncus* genus and unique viruses present in ME/CFS microbiomes. This study therefore provides a framework and rationale for studies of larger cohorts of patients to further investigate disease-associated interactions between the intestinal virome and the bacteriome.

## 1. Introduction

The gastrointestinal (GI) virome [1,2] comprises eukaryotic viruses, bacteriophages (or phages) [3], archaeal viruses, endogenous retroviruses and other rare viruses from diverse environmental sources including water, animals and plants [4]. Phages account for the vast majority (>90%) of the intestinal virobiota [5,6] with the capability of changing (gaining or losing) the structural and functional compositions of the microbial community, thereby contributing to intestinal homeostasis and the health of the human host [7]. Reductions in the richness and diversity of the human intestinal virome have been described in patients with chronic inflammatory GI disorders such as inflammatory bowel disease (IBD) [1], suggesting that an alteration of the intestinal virome may contribute to the pathophysiology of chronic diseases.

Virome studies are constrained by the high genetic diversity and lack of universally conserved genes amongst viruses. Among culture-independent methods, metagenomics, which enables the composition of the collective microbial genomes within an environmental sample to be resolved, is currently the most reliable way to survey the human virome. Commonly, for human GI virome studies, this approach is applied to either the metagenomes from fractionated faecal samples enriched in virus-like particles (VLP) or from whole metagenome sequences (WMSs) of bulk, non-fractionated, samples. Individually these approaches yield incomplete virus profiles with VLP-based metagenomes, for example, failing to account for prophages that can be detected by WMS-based approaches.

To address this shortcoming, we have combined DNA viromes from isolated VLPs and whole metagenomes from the same faecal sample using a small test cohort of healthy individuals and patients with severe myalgic encephalomyelitis/chronic fatigue syndrome (ME/CFS). ME/CFS is a severe disabling and debilitating heterogeneous disorder affecting multiple organ systems and producing symptoms including myalgia, post-exertional malaise (PEM), sleep disturbance, GI dysfunction, neurological and cognitive impairment and prolonged unexplained fatigue [8,9]. Approximately 25% of patients are severely affected, being housebound or bedbound for prolonged periods of time, requiring in-home assistance, often provided by family members [10].

The aetiology of ME/CFS remains unknown with several hypotheses being proposed including chronic viral infections [11,12], intestinal microbial dysbiosis [13,14,15,16], metabolic disorders [17,18], mitochondrial dysfunction [19,20] and/or autoimmunity [21,22,23]. Chronic viral infection is supported by clinical- and laboratory-based observations indicating that most ME/CFS cases (50–80%) are associated with and may follow prolonged influenza-like symptoms including post-viral fatigue [24,25]. Several human eukaryotic viruses are potential aetiological agents of ME/CFS, including human herpesviruses 6 (HHV-6) and HHV-7 [26], *Epstein–Barr virus* (EBV) and *Cytomegalovirus* (CMV) [27,28], *Parvovirus B19* [29,30,31] and enteroviruses [32,33]. A potential source of infective and pathogenic viruses is the human virome and, in particular, the GI virome.

The possibility that GI viral infection contributes to the pathology of ME/CFS is supported by clinical- and laboratory-based observations [29,33,34]. Most ME/CFS patients (92%) have persistent or intermittent symptoms of GI dysfunction and suffer from irritable bowel syndrome (IBS) [35], with associated alterations in the intestinal virome [1,2,36]. In one of the few virome studies of ME/CFS patients carried out to date, alterations in the intestinal (faecal) virome have been identified in a pair of identical twins discordant for ME/CFS and living apart from each other for several years, with the affected twin displaying a comparative increase in the abundance of phages of the class *Caudoviricetes*, of which the majority were predicted siphoviruses and myoviruses [34].

Here we set out to determine if, by combining and comparing two datasets derived from VLP-enriched metagenomes and WMSs from stool samples obtained from severely-affected ME/CFS patients and healthy individuals living in the same environment and household that care for the patients (i.e., same household healthy controls; SHHCs), it is possible to obtain more high-quality viral genomes. The datasets were used to identify any systematic differences in the virome or bacteriome ME/CFS patients versus healthy individuals, the nature of any differences in viromes obtained using different viral isolation methodologies, and if it is possible to predict disease-associated bacterial hosts for the viruses uniquely present in ME/CFS patients.

## 2. Results

### 2.1. Study Cohort and VLP Characterisation

Nine severely affected ME/CFS patients who met the 2003 Canadian Consensus Criteria and The National Institute for Health and Care Excellence (NICE) 2007/CG53 guidelines were recruited from the CFS Service of the Epsom and St. Helier NHS Foundation Trust University Hospital or the East Coast Community Healthcare Centre ME/CFS Service along with eight SHHCs who were parents and/or caregivers of these patients (Table 1). ME/CFS patients were all female, with an average age of 33.8 ± 13 years (mean ± SD), consistent with the known higher prevalence of ME/CFS in females than males in the general population. Amongst these, four severely affected patients had a self-reported GI disorder. The age of onset and duration of illness varied among patients. All patients had received their diagnosis at least six months prior to recruitment to the study. SHHC subjects comprised three males and five females of average age 53.6 ± 15.5 years (mean ± SD, *n* = 8). No SHHCs reported any GI disorders or symptoms at the time of enrolment or sample collection.

Transmission electron micrographs (TEM) of VLP samples (*n* = 17) (Figure 1) showed a predominance of phages, particularly tailed double-stranded DNA (dsDNA) phages belonging to the class *Caudoviricetes* (including three common morphotypes: siphoviruses, myoviruses, and podoviruses), alongside a filamentous virus seen in one sample (Figure 1E). SYBR Gold staining and epifluorescence microscopy (EFM) were used to quantify enriched VLPs, which numbered 1.6 × 10^9^ ± 8.1 × 10^8^ VLP/g faeces per sample (mean ± SD), with faecal VLPs yielding 2518.5 ± 1832.9 ng DNA per sample (mean ± SD) (Appendix A). There was no difference in the number (mean ± SD) of ME/CFS-derived VLPs (1.6 × 10^9^ ± 9.3 × 10^8^ VLP/g faeces; *n* = 9) compared with SHHC-derived VLPs (1.5 × 10^9^ ± 7.1 × 10^8^ VLP/g faeces; *n* = 8). However, ME/CFS-derived VLPs yielded less DNA (mean ± SD) than SHHC-derived VLPs (1585.2 ± 1411.0 ng DNA versus 3568.4 ± 1738.8 ng DNA, *p* < 0.05, two-tailed *t* test) (Appendix A).

Genomic DNA from bulk faecal samples from seven ME/CFS and five SHHC patients was used to profile non-enriched viral and prokaryotic genomes using whole metagenomic sequencing (WMS). Five faecal samples contained insufficient biomass for parallel WMS and VLP analyses and were therefore prioritised for VLP enrichment. Three sequence datasets were created: (1) DNA viromes derived from VLP-enriched metagenomes; (2) DNA viromes derived from WMSs; and (3) bacteriomes derived from whole metagenomes (WMS-Bac).

### 2.2. Quantitative and Qualitative Assessment of Enriched VLP- and WMS-Derived Viral Genomes

The bioinformatic pipeline used for analysing cohort-associated viromes and bacteriomes is depicted in Figure 2. Statistical analyses of both the VLP and WMS datasets are detailed in Appendix A.

From the enriched VLP dataset, 311,208 uncultivated viral genomes (UViGs) were detected. These UViGs were clustered and dereplicated at 95% average nucleotide identity (ANI) over 85% of the contig length to generate 192,146 primary non-redundant viral operational taxonomic units (vOTUs). From these 1231 non-redundant, high-quality (HQ) vOTUs were retained after removal of genome fragments of less than 1 Kbp in length; low-quality (<50% genome completeness) genomes, and undetermined genomes; (HQ) vOTUs included 193 complete (62,030.8 ± 40,950.2 bp, mean ± SD), 488 high-quality (>90% completeness) and 550 medium-quality (50–90% completeness) genomes, with an average length of 44,581.9 ± 33,085.6 bp per vOTU (mean ± SD) (Appendix A). In the WMS dataset, 688 non-redundant HQ-vOTUs, including 63 complete genomes (90,560.9 ± 63,003.9 bp, mean ± SD), were retained, with an average length of 49,477.2 ± 41,680.2 bp per vOTU (mean ± SD) (Appendix A). To assess the similarity between the VLP and WMS datasets, we aligned 688 WMS-HQ-vOTUs against 1231 VLP-HQ-vOTUs, resulting in 600 non-redundant sequence alignments with the best hit being obtained. Of these, 193 alignments were highly likely to be identical between both datasets with high reliability, as determined using several critical thresholds (Appendix A).

Moreover, our data showed that the putative integrated viruses (proviruses or prophages) detected in the WMS-HQ-vOTUs outnumbered those detected in the VLP-HQ-vOTUs (Table 2, Appendix A). Analysis of plasmid sequences showed that the number of plasmids was reduced in both datasets after removing low-quality and undetermined genomes, with very low plasmid contamination retained in both datasets, as indicated by the low number of plasmid hallmark genes and low marker enrichment scores (Table 2, Appendix A).

Using the same VLP and DNA isolation protocol three process-control samples (i.e., TBT buffer alone with no stool sample spiking [designated as Negative Controls (NC)] generated very low numbers of non-human sequences (Appendix A). For taxonomic classification, NC-derived reads were primarily assigned to members of the phyla *Proteobacteria*, *Actinobacteria*, *Bacteroidetes*, and *Firmicutes*. Only 0.48% of reads in NC1 (859/180,120), 0.01% in NC2 (1/9296), and 1.16% in NC3 (647/55,832) were assigned to viruses such as *Lactococcus* phages, pahexaviruses, moineauviruses, and human polyomaviruses (Appendix A). In addition, only 6–15% of the NC-derived reads could be mapped to VLP-HQ-vOTUs, with very low coverage observed after VLP enrichment (Appendix A). No HQ-vOTUs were present in the process-control samples using the assembly and quality filtering workflow described above.

### 2.3. Cluster Analysis and Taxonomic Assignment of the HQ-vOTUs

To assess the distribution of related groups of viruses in the VLP dataset, 37 viral clusters (VCs) were uniquely composed of ME/CFS-associated vOTUs compared to 10 VCs uniquely associated with SHHC vOTUs; 147 shared VCs were identified that consisted of communal virus groups present in both samples (Figure 3, Appendix A).

From the WMS dataset, 17 unique VCs were ME/CFS-associated and 10 were SHHC-associated; there were 93 shared VCs composed of communal virus groups (Figure 4, Appendix A). The distributions of HQ-vOTUs and VCs in the WMS dataset were different from those in the VLP dataset, reflecting different viral compositions obtained from both datasets.

Most VLP-HQ-vOTUs were grouped into the classes *Caudoviricetes*, *Faserviricetes* and *Malgrandaviricetes*, with the remainder being unknown or unclassified. We also identified filamentous ssDNA phages that are members of the family *Inoviridae*. Similarly, WMS-HQ-vOTUs with a ‘clustered’ or ‘clustered/singleton’ status were grouped into the classes *Caudoviricetes*, *Faserviricetes* and *Malgrandaviricetes*, with the remainder being unknown or unclassified. Of note, from the VLP dataset, we identified six VCs with ‘clustered’ or ‘clustered/singleton’ vOTUs that contained human-associated *Crassvirales* phages. From the WMS dataset, only four VCs composed of ‘clustered’ or ‘clustered/singleton’ vOTUs were associated with the *Crassvirales* phages (Appendix A).

### 2.4. Detection of GI Eukaryotic DNA Viruses

To assess the presence of eukaryotic DNA viromes in our datasets, mapped viral reads were first retrieved from VLP and WMS HQ-vOTUs. Overall, very few mapped viral reads (<1%) were classified as human-associated eukaryotic viruses with a majority being unclassified or phages (Appendix A). Of the classified viral reads, some unique human-associated eukaryotic viruses were seen in the ME/CFS-derived samples including human betaherpesvirus 6A (HHV-6A), papillomaviruses, coronavirus NL63 and adenovirus 54, with others being communal viruses shared in both ME/CFS- and SHHC-derived samples (Appendix A). Next, we undertook a more stringent analysis using VIRify for the detection of eukaryotic viruses from the unknown or unclassified vOTUs. No eukaryotic viruses were detected in either the VLP or WMS datasets, with a majority remaining unknown or unclassified (Appendix A).

### 2.5. Macrodiversity of Intestinal Viromes

With respect to the beta diversity of intestinal DNA phages, we found that within the same sample, there were two distinct clusters represented by VLP- and WMS-derived groups, respectively (*p* < 0.001) (Figure 5A). There was no evidence of clustering by ME/CFS status (*p* = 0.691) with the viral composition of the SHHC sample being similar to that of the corresponding ME/CFS sample (PERMANOVA, *p* = 0.172) (Figure 5B). We also determined the mean relative abundance of intestinal viruses across ME/CFS- and SHHC-derived samples (Figure 5C) and measured the relative virus abundance for each individual sample separated by household-matched pairs at the family and genus levels (Appendix A). Overall, the most dominant viral families across the samples belonged to the class *Caudoviricetes* comprising tailed phages (Figure 5C). Moreover, the Shannon index of alpha diversity was higher in the WMS-derived datasets than in the VLP-enriched datasets (difference = 2.28, *p* < 0.0001) (Figure 5D). Although there was no evidence of any differences between ME/CFS- and SHHC-derived samples, when comparing VLP-enriched and WMS-derived datasets, there were differences in the relative abundance of viruses from several families (Figure 5E). Enriched VLPs had higher abundances (as measured by the centred log-ratio, CLR) in the families *Suoliviridae* (*p* < 0.0001), *Salasmaviridae* (*p* < 0.0001), *Intestiviridae* (*p* < 0.0001), *Steigviridae* (*p* = 0.002) and *Crevaviridae* (*p* < 0.001), and lower abundances in *Duneviridae* (*p* < 0.0001), *Winoviridae* (*p* < 0.0001) and other unclassified families (*p* < 0.05) (Figure 5E).

### 2.6. Macrodiversity of Intestinal Bacteriomes

To assess the impact of ME/CFS on intestinal bacterial communities, we first determined the beta diversity at the bacterial genus level (Figure 6A). Using Jaccard distance to assess species similarity, we found that within each household-matched pair, the SHHC subjects had bacterial compositions similar to those of the corresponding ME/CFS patient (*p* = 0.002), with no evidence of any differences between the ME/CFS and SHHC groups (*p* = 0.924) (Figure 6A). Overall, across the WMS-derived datasets, the three most abundant bacterial phyla were *Firmicutes* (46%), *Bacteroidetes* (25%), and *Actinobacteria* (2.7%) (Appendix A). Among the 12 most abundant bacteria at the genus level, *Alistipes* (8.2%), *Bacteroides* (4.3%) and *Faecalibacterium* (4%) were prominent (unknown genera were excluded; Figure 6B). *Prevotella* dominated the profile of one individual (sample 2C) (Figure 6B). Comparison of the centred log-ratios (CLRs) among the 12 most abundant bacterial genera showed no significant differences between the ME/CFS and SHHC groups (Appendix A). The ‘Observed richness’, ‘Chao1′ and ‘Shannon’ indices were used to determine the alpha diversity of intestinal bacteria (Figure 6C,D). Overall, no significant differences were observed between bacteria from the ME/CFS- or SHHC-derived samples for any measure of observed species richness (*p* = 0.877), Chao1 (*p* = 0.828), and Shannon index (*p* = 0.975), although there was a high intraclass correlation within the household-matched pairs for the observed richness and Chao1 measures (ICC = 0.754 and 0.837, respectively).

Finally, using data on between- and within-household variations in the relative abundance of bacterial genera, we performed a power calculation to estimate the number of samples needed in a future study to reliably identify the differential abundance of individual genera between groups. The data suggested that at least 15 household-matched pairs would be required to detect ‘true’ differences on the scale of a 10-fold difference in the abundance of specific bacterial genera in matched pairs, with a 90% statistical power at a critical threshold of *p* < 0.001, whereas more than 100 pairs would be needed to detect 90% of genera that had a true two-fold change in abundance between groups. (Appendix A).

### 2.7. Predicting Disease-Associated Bacterial Hosts for the HQ-vOTUs

Using a combination of 319 metagenome-assembled genomes (MAGs) from this study and reference bacterial genomes, 81 shared bacterial hosts were predicted in the VLP- and WMS-derived datasets, 94 hosts were uniquely predicted in the VLP dataset, and 25 hosts were uniquely predicted in the WMS dataset (Figure 7, Appendix A). Next, we investigated whether these predicted hosts were ME/CFS- or health-associated by interacting with either ME/CFS- or SHHC-unique viruses, or communal viruses shared in both samples (Appendix A). Of the 81 bacterial hosts shared in both the VLP and WMS-derived datasets, 11 bacteria including the genera QAMI01, *Erysipelatoclostridium*, *Lacticaseibacillus*, Firm-11, *Christensenella*, *Eubacterium*, *Anaerobutyricum*, *Anaerostipes*, *Acutalibacter*, *Veillonella* and *Salmonella* potentially associated with ME/CFS-unique viruses (Appendix A). Of the 94 VLP-derived bacterial hosts, 37 bacteria (*Clostridium*_J, *Clostridium*_AP and *Anaerotruncus*) were predicted to be associated with ME/CFS-unique viruses. In addition, only six WMS-derived bacterial genera (*Bacteroides*_G, *Campylobacter*_D, HGM12998, *Pseudoflavonifractor*, *Oxalobacter* and *Klebsiella*) were potentially associated with ME/CFS-unique viruses (Appendix A).

Finally, we compared the 20 most abundant bacterial genera from ME/CFS samples with host predictions to identify possible ME/CFS-associated bacterial hosts (Table 3). Within the 20 most abundant genera, most bacterial hosts such as *Faecalibacterium* and *Roseburia* were associated with communal viruses. Although the genera *Bacteroides*, *Ruminococcus*, *Clostridium* and *Eubacterium* were associated with diverse viruses, including a mixture of communal viruses and ME/CFS- and SHHC-unique viruses, the genus *Anaerotruncus* from the VLP-derived dataset was the only bacterium associated with ME/CFS-unique viruses with no identifiable interactions with SHHC-derived or communal viruses (one unknown virus and two unclassified *Caudoviricetes* phages) (Table 3, Appendix A).

## 3. Discussion

By combining VLP and WMS viromes, we obtained a more complete and comprehensive profile of the human GI virome using a small test cohort of severely affected ME/CFS patients and unaffected healthy individuals living in the same household as the patients. We also predicted ME/CFS-associated bacterial hosts that interact with unique viruses in severely affected ME/CFS patients, which may provide new insights into GI dysbiosis in ME/CFS.

An important feature of our study design is the inclusion of SHHC subjects to account for and minimise any impact of environmental factors and the indoor environment on the microbiome (prokaryome) [37]. Individuals sharing the same household have similar microbial compositions compared to those living in different households, thereby increasing confidence in attributing differences in microbiomes in ME/CFS patients to the disease itself [37]. While SHHCs co-habit and live in the same environment as the patient, it is important to acknowledge differences in sex (9/9 female patients, 5/8 female SHHCs) and age (average of 33.8 years old for patients and 53.6 years old for SHHCs) that could potentially contribute to variations in individual microbiomes [38,39].

We recovered ~10^9^ VLP/g faeces per sample after VLP enrichment, with the most common viral morphotypes being tailed phages, which is broadly consistent with other studies [40,41,42,43]. Non-human reads derived from VLP enrichment had a higher mapping rate and average read coverage to the sorted high-quality (HQ) vOTUs than those derived from WMS, which is consistent with most mapped reads being derived from viruses after VLP enrichment. Moreover, non-human reads derived from the negative control samples had very low or no mapping rates and coverage of VLP-HQ-vOTUs, suggesting low levels of contamination in our enriched virome dataset. The HQ-vOTU datasets from enriched VLPs and WMSs from the same stool sample overlapped but had distinct components. Most enriched viruses are likely extracellular lytic viruses such as crAss-like phages that would potentially exhibit typical predator–prey dynamics [44]. However, in general, extracellular lytic phages can be sparse and infrequent within the adult GIT, with the exception of lytic *Crassvirales* and *Microviridae* phages that stably reside in the GIT of healthy adults [45]. Many VLP-enriched enteric viruses are likely temperate phages that enter the lytic cycle [46,47]. This could explain why the number of proviruses in the VLP-derived samples was lower than that in the WMS-derived samples. In contrast, most viruses detected in WMSs are more likely to be dormant prophages within bacterial genomes, with prophage induction being associated with GI inflammation and dysbiosis [48,49,50]. This suggests that induced or non-induced prophages can play an important role in modulating GI microbial ecology. Using both VLP- and WMS-based isolation methods for sequencing is advantageous, given the dominance of temperate phages or induced prophages interacting with GI microbiota through various mechanisms and dynamics [46,49,51]. Together, VLP- and WMS-based isolation methods provide complementary datasets that include more diverse, high-quality, and larger complete viral genomes comprising extracellular viruses and integrated proviruses, which provides a more comprehensive profile of the human intestinal DNA virome, while also capturing a small overlap (~8.5%) in viral populations [6,45].

Although most VCs were composed of communal viruses shared by both ME/CFS- and SHHC-derived samples, we identified specific VCs comprising unique viruses present only in ME/CFS- or SHHC-derived samples. VCs and viruses unique to ME/CFS-derived samples are likely to be associated with the disease. Of the classified VCs, most were *Caudoviricetes* phages, including members of human-associated crAss-like phages, with a minority being members of the *Microviridae* and *Inoviridae* families (i.e., ssDNA viruses) and others. While members of *Microviridae* have been considered predominant members of the human GI virome [45], they may be overrepresented in virome studies using multiple displacement amplification (MDA) due to biases that preferentially amplify short circular ssDNA [52]. To avoid MDA-based amplification biases and overestimation of ssDNA viruses, we used a conventional PCR method for library preparation prior to sequencing. Few bacterial and viral taxa were detected in the process-control samples which are likely due to amplification or sequencing bias or contamination, with no HQ-vOTUs being present in the process-control samples as a result of our use of stringent sorting criteria and approaches. Moreover, we used a chloroform-free protocol for VLP isolation to avoid the depletion of enveloped viruses. Members of the order *Crassvirales* are the most abundant tailed dsDNA phages within the human GIT [53,54]. However, despite using vConTACT 2 alongside the recently updated reference viral database, many VCs/vOTUs still remain unknown or unclassified.

To identify the presence of human GI eukaryotic DNA viromes in our samples we initially retrieved mapped viral reads from the VLP and WMS HQ-vOTUs. However, very few viral reads have been classified as human-associated eukaryotic viruses. The majority were phages or unclassified, with very few reads identifying unique eukaryotic DNA viruses in the ME/CFS-derived samples, which included HHV-6A that has a known association with ME/CFS [26]. Due to the current limitations of vConTACT 2 [55], no eukaryotic DNA viruses were classified in our cluster analysis. Furthermore, using a more stringent approach and VIRify to detect eukaryotic viruses from the unknown or unclassified VLP and WMS HQ-vOTU groups, no HQ-vOTUs were identified as eukaryotic virus with the remainder being unclassified or DNA phages. Therefore, we cannot exclude the possibility that eukaryotic DNA viruses were present in our samples but could not be detected using the available tools. In addition, no RNA viruses or phages were detected because the extraction protocols used were not optimised for RNA viruses/phages. Although some human-associated pathogenic GI viruses are RNA viruses (e.g., rotavirus, norovirus and enteroviruses), other enteric RNA phages may originate from plant-based food [56]. Therefore, future GI virome studies should include enteric RNA viruses.

Considering the limited sample sizes of this study, it is perhaps unsurprising that no significant differences were observed in viral abundance and diversity between the ME/CFS and SHHC groups. Consistent with recent ME/CFS microbiome studies [13,16], our data indicated that *Firmicutes*, *Bacteroidetes* and *Actinobacteria* were the three most dominant phyla in the intestinal microbiota, with our data also showing an intraclass correlation within the household-matched pairs across many bacterial genera. Although no statistical evidence is presented for these differences, the intestinal microbiome of severely affected ME/CFS patients may be associated with a reduction in *Firmicutes* and an increase in *Bacteroidetes* [13,16,57]. There were no quantitative differences in the alpha and beta diversities of bacteria between the ME/CFS and healthy control samples, which may reflect the limited sample sizes and, with our statistical power calculation identifying a requirement of more than 100 pairs to detect 90% of genera with a true two-fold change in abundance between the groups.

Phages can modulate the dynamics of microbial communities in human GIT via direct or indirect trans-kingdom interactions [2,58]. Using an in silico approach we predicted unique and shared bacterial hosts from the VLP and WMS datasets. Among the 20 most abundant ME/CFS-derived bacteria, the genera *Bacteroides*, *Ruminococcus*, *Clostridium*, *Anaerotruncus* and *Eubacterium* were associated with ME/CFS-unique viruses derived from patient samples; of these, most bacteria were short-chain fatty acid (SCFA) producers. Recent reports have identified that reductions in butyrate-producing bacteria, such as *Faecalibacterium prausnitzii*, *Roseburia* spp. and *Eubacterium Rectale*, are associated with butyrate deficiency and disease severity in ME/CFS [14,59]. An increased abundance of the genus *Bacteroides* has also been observed in ME/CFS patients compared to non-affected healthy individuals [13]. Moreover, most of the abundant bacterial hosts were broadly associated with a variety of communal viruses or ME/CFS- or SHHC-unique viruses. Of note, *Anaerotruncus* derived from VLP-enriched samples is uniquely associated with ME/CFS-unique viruses supporting the previously reported increase in abundance of *Anaerotruncus* genera in faecal metagenomes of a cohort of ME/CFS patients [60]. The abundance of the *Anaerotruncus* genus, which may comprise as few as three species (*A. colihominis*, *A. massiliensis* and *A. rubiiinfantis* [61,62,63]), is associated with higher bacterial gene richness in the GIT [64] and is presumed to be anti-inflammatory [65]. Consistent with this prediction are inverse correlations with *A. colihominis* abundance and high BMI and elevated serum triglycerides [66] and cognitive function scores in Alzheimer’s patients [67]. In addition, in animal models of multiple sclerosis *A. colihominis* is associated with the induction of regulatory (RoRγt^+^) T cells and amelioration of disease [68] and is a constituent of a therapeutic consortia of 17 Clostridia strains derived from human stool [69]. Thus, we can speculate that alterations in members of the genus *Anaerotruncus* and their host-specific phages are potentially associated with the pathology of ME/CFS, which warrants further investigation in larger patient cohorts.

Collectively, these findings suggest that disease-associated viruses may by directly or indirectly modulating their bacterial hosts contribute to dysbiosis in ME/CFS.

## 4. Materials and Methods

### 4.1. Study Participants and Sample Collection

All participants were registered with the CFS service of the St. Helier Hospital, Surrey, UK, or the ME/CFS Service of the East Coast Community Healthcare Centre (ECCHC), Norfolk/Suffolk, UK, and were recruited to the study between 2017 and 2019. All fulfilled the Canadian [70] and NICE (National Institute For Health And Care Excellence) 2007/CG53 guideline [71] diagnostic criteria of CFS, which were used alongside clinical history and the Hospital Anxiety Depression Scale (HADS) [72] to exclude patients with significant clinical depression and anxiety. Participants consuming probiotic capsules or antibiotics within four weeks prior to sample collection were also excluded from the study. Disease status was based on the following: (1) few or no activities of daily living, (2) severe cognitive difficulties, (3) wheelchair dependency for mobility, (4) unable or rarely able to leave the house or are bed-bound, requiring assistance with washing, toilet use and feeding, and (5) often displaying significant worsening symptoms with mental or physical exertion, and in extreme cases unable to tolerate noise and are light sensitive. The inclusion criteria for SHHC recruitment were as follows: (1) the individual had to be living with or caring for the patient, (2) men or women aged between 18 and 70 years, (3) no current or ongoing medical conditions, and (4) able to provide informed consent. SHHC participants with long-term medical conditions (e.g., inflammatory bowel disease, IBS), suffering from autoimmune diseases, significant anxiety or depression, taking immunomodulatory drugs, statins, beta-blockers, or steroids, and consuming probiotic capsules or antibiotics within four weeks prior to sample collection were excluded from the study. Ethical approval was obtained from the University of East Anglia Faculty of Medicine and Health Sciences Research Ethics Committee in 2014 (reference FMH20142015–28) and the Health Research Authority NRES Committee London Hampstead (reference 17/LO/1102; IRAS project ID: 218545). Subjects recruited in 2017 also completed a shortened SF-36, the Chalder fatigue questionnaire, a self-efficacy questionnaire, visual analogue pain rating scale and Epworth sleepiness scale. Written informed consent was obtained from all participants.

Faecal samples were collected from the participants’ homes, followed by transportation in a chilled and insulated Fecotainer^®^ (Excretas Medical BV, Enschede, The Netherlands) for delivery to the laboratory within 6 h of collection. Upon receipt, the samples were aliquoted and stored at −80 °C prior to analysis.

### 4.2. Faecal VLP and VLP DNA Isolation

The faecal VLP and VLP DNA isolation protocol has been described previously [73]. To evaluate the recovery efficiency of VLP isolation, two preliminary spiking-and-recovery assays were performed prior to formal experiments (Appendix A). For VLP and DNA isolation, frozen faecal aliquots (3–4 g faeces per aliquot) were homogenised in sterile TBT buffer (100 mM Tris-HCl, pH 8.0; 100 mM NaCl; 10 mM MgCl_2_·6H_2_O) by vortexing and then incubated on ice for 1 h. The faecal homogenates were then centrifuged at 11,200× *g* for 30 min at 10 °C for two rounds. Supernatants were filtered sequentially through 0.8 µm (Sterlitech, Auburn, WA, USA) and 0.45 µm (Starlab Ltd., Milton Keynes, UK) polyethersulfone cartridge filters. NaCl (final concentration 6%, *w*/*v*; Sigma-Aldrich Ltd., Gillingham, UK) was then added to faecal filtrates and mixed, followed by the addition of PEG 8000 (final concentration 10%, *w*/*v*; Sigma-Aldrich Ltd., Gillingham, UK). After resuspension, the samples were incubated at 4 °C for 16 h. PEG-precipitated VLPs were harvested by centrifugation at 4500× *g* for 60 min at 4 °C and VLP-containing pellets were resuspended in ~500 μL of TBT buffer. To avoid loss of enveloped viruses, no chloroform was included in the protocol. VLP suspensions were then treated with 10 U TURBO DNase (Invitrogen/Thermo Fisher Scientific, Hemel Hempstead, UK) and 20 U RNase I (Ambion/Thermo Fisher Scientific, Hemel Hempstead, UK) at 37 °C for 45 min. EDTA (final concentration 15 mM, pH 8.0; Invitrogen/Thermo Fisher Scientific, Hemel Hempstead, UK) was then added to stop the reaction, followed by heat inactivation at 75 °C for 10 min. Proteinase K (100 μg; Ambion/Thermo Fisher Scientific, Hemel Hempstead, UK) and 5% (*w*/*v*) SDS (final concentration 0.5%, *w*/*v*; Sigma-Aldrich Ltd., Gillingham, UK) were added and incubated at 56 °C for 75 min, followed by the addition of lysis buffer (final concentration 133.3 mM Tris-HCl, pH 8.0; 33.3 mM EDTA, pH 8.0; 3.3% SDS, *w*/*v*) and incubation at 65 °C for 15 min. The VLP lysate was treated with an equal volume of phenol/chloroform/isoamyl alcohol (25:24:1, *v*/*v*/*v*; Thermo Fisher Scientific, Hemel Hempstead, UK) and mixed thoroughly by vortexing for 30 s, followed by two rounds of centrifugation at 15,000× *g* for 5 min at ambient temperature. The resulting aqueous phase was collected and transferred to a ZR genomic DNA Clean & Concentrator^™^-25 column (Zymo Research; Cambridge Bioscience Ltd., Cambridge, UK) with purified DNA eluted in low-EDTA elusion buffer. To increase DNA concentrations, 10–12 DNA aliquots isolated from each sample (i.e., average amounts of faeces: 30.9 ± 11.8 g per sample, mean ± SD, *n* = 17) were pooled and concentrated to a final volume of 60–100 µL (Appendix A). The concentrated DNA was stored at −80 °C. Using the same VLP and DNA isolation protocol, three negative control (NC) samples (TBT buffer only with no spiking with stool sample) were included as process controls to monitor contaminations and false-positive signals in high-throughput sequencing. These NC samples (NC1–3) were added during the homogenisation step and throughout the process of isolation, library preparation, and sequencing. The quantity and quality of the recovered VLP DNA were determined using the Nanodrop and Qubit^™^ 1× dsDNA HS Assay Kit (Thermo Fisher Scientific, Hemel Hempstead, UK).

### 4.3. Epifluorescence Microscopy (EFM)-Based Enumeration of Enriched Faecal VLPs

The primary Invitrogen^™^ SYBR^™^ Gold stock solution (concentration 10,000×; Thermo Fisher Scientific, Hemel Hempstead, UK) was diluted to a working concentration of 0.25% (*v*/*v*) with sterile Ambion^™^ nuclease-free water (Thermo Fisher Scientific, Hemel Hempstead, UK) and stored at −20 °C until used. For use, SYBR Gold solution was defrosted at ambient temperature in the dark for 15 min prior to use. A total of 20 µL of PEG-precipitated VLPs were diluted in 900 µL of nuclease-free water to which 100 µL of SYBR Gold working solution (final concentration 0.025%, *v*/*v*) was added, mixed gently, then incubated in the dark for 15 min at ambient temperature. An Omnipore 0.45 µm, 13 mm PTFE backing filter (Millipore/Sigma-Aldrich Ltd., Gillingham, UK) was placed on top of a Swinnex filter holder with a silicone gasket (Millipore/Sigma-Aldrich Ltd., Gillingham, UK). The backing filter was rinsed by nuclease-free water followed by placing a 0.02 µm white Whatman^™^ Anodisc 13 mm filter membrane (Sigma-Aldrich Ltd., Gillingham, UK) on top of the backing filter. The Anodisc filter membrane was rinsed by nuclease-free water using a low-vacuum pressure (~20 kPa; Millivac-Maxi vacuum pump, Millipore Ltd., Livingston, UK), as described previously [40,74]. The SYBR Gold-labelled VLP sample (20 µL) was then fixed on the Anodisc membrane using low-vacuum pressure until all liquid had passed through the membrane. The membrane was washed with 1 mL of nuclease-free water to remove excess dye. The filter membrane was then transferred to a Whatman^®^ filter paper disc (Sigma-Aldrich Ltd., Gillingham, UK) and left to dry for 1 min. Prior to adding a coverslip, Fluoromount-G^®^ antifade mounting reagent (SouthernBiotech, Birmingham, AL, USA) was spotted on a microscope slide and the dried filter membrane was then placed onto the mounted droplet. The slide was left in the dark at ambient temperature for 16 h and slides were subsequently imaged using a Zeiss Axio Imager M2 widefield epifluorescence microscope (Carl Zeiss Microscopy Ltd., Cambridge, UK) with Alexa Fluor 488 channel and 100× oil objective lens (Carl Zeiss Microscopy Ltd., Cambridge, UK). For each slide, 20 digital images were captured, and SYBR Gold-labelled viral particles were viewed and counted using ImageJ (v1.52p) software (https://imagej.net/, accessed on 8 August 2019) with the average number of VLPs per field multiplied by sample dilution factor and microscope conversion factor (i.e., area of 13 mm Anodisc filter/area of field of view) and then divided by sample volume [74]. Additionally, negative control samples were included as process controls to monitor contamination, including non-stained VLP suspension, dye incubated with sterile TBT buffer and dye incubated with nuclease-free water. One VLP sample was used as positive control for each batch of staining and imaging to monitor experimental consistency.

### 4.4. Transmission Electron Microscopy (TEM)

Five microliters of diluted faecal filtrate was applied to carbon-film on copper 400 mesh grids (EM Resolutions Ltd., Keele, UK) for 1 min and excess liquid removed by wicking the edge of the grid with Whatman filter paper, followed by 2 min incubation with 0.5% (*w*/*v*) uranyl acetate (UA) solution (BDH/VWR International Ltd., Lutterworth, UK). Excess UA was then removed by wicking with filter paper. Each grid was vapor fixed by adding 1 mL of 2.5% (*v*/*v*) glutaraldehyde to the dish containing the dried grids for a minimum of 2 h. Imaging was then performed using a Talos F200C TEM microscope (Thermo Fisher Scientific, Hemel Hempstead, UK) at 200 kV with a Gatan OneView digital camera (AMETEK GB Ltd., Leicester, UK).

### 4.5. Library Preparation and Whole Metagenomic Sequencing of Enriched VLP DNA

VLP DNA samples collected from 17 faecal samples along with three negative control samples were constructed in a pooled PCR-based barcoding library by the Quadram Institute Bioscience in-house sequencing service using the Nextera XT DNA Library Preparation Kit (Illumina Inc., Cambridge, UK). DNA samples were normalised to 0.5 ng/µL with 10 mM Tris-HCl (pH 8.0) prior to library preparation, followed by tagmentation and adapter ligation using 1 ng of the input DNA template. Input DNA was randomly fragmented with engineered Tn5 transposase generating a mean of ~300 bp of DNA inserts with adapters, followed by PCR amplification to barcode the adapter-ligated DNA input. The PCR programme was 72 °C for 3 min and 95 °C for 1 min, followed by 14 cycles of 95 °C for 10 s, 55 °C for 20 s, and 72 °C for 3 min. All libraries were then pooled, cleaned with KAPA pure beads (Roche Diagnostics Ltd., Hassocks, UK), and quantified using an Invitrogen^™^ Quant-iT dsDNA assay kit (Thermo Fisher Scientific, Hemel Hempstead, UK). A pooled library sample was then sequenced using 2 × 150 bp paired-end chemistry (PE150) on an Illumina HiSeq X Ten platform (Novogene Ltd., Cambridge, UK). The raw sequencing data of ME/CFS- and SHHC-derived samples had a Q30 score of >90% generating an average depth of 14 Gb per sample. In total, 1,586,595,530 paired-end raw sequencing reads (93,329,148.8 ± 48,857,207.3 per sample, mean ± SD, *n* = 17) were obtained. Three negative control samples (NC1–3) were included as process controls and had a Q30 score of >88% with approximately 0.1 Gb of output per sample generating 791,934 paired-end raw sequencing reads (263,978 ± 307,643.7, mean ± SD, *n* = 3). Paired-end sequencing reads were provided in FASTQ format. All raw sequencing reads were pre-processed to trim and remove adapters, low quality (Q-value ≤ 38) and N nucleotides using readfq [75] and FxTools (v0.17) [76] performed by Novogene Ltd. (Cambridge, UK). Human genomic DNA detected by Kraken 2 (v2.0.8) [77,78] against the Genome Reference Consortium Human Build 37 (GRCh37/hg19) database was removed using the confidence at 0.5, followed by further cleaning of the reads using fastp (v0.23.1) [79] with a quality cut-off of 20 prior to genome assembly. Kraken 2 was also used to taxonomically classify the sequences of the negative controls (NC1–3), and the sequences were considered biases or contaminants.

### 4.6. Faecal Genomic DNA Isolation, Library Preparation and Whole Metagenomic Sequencing

As five faecal samples contained insufficient biomass for both whole-metagenomic shotgun (WMS) sequencing and VLP analysis, they were prioritised for VLP enrichment. The remaining 12 faecal samples (i.e., seven ME/CFS and five SHHC samples) were processed for WMS sequencing. Briefly, genomic DNA was extracted from ~250 mg aliquot per sample using FastDNA^™^ SPIN Kit for Soil (MP Biomedicals, Eschwege, Germany), following the manufacturer’s instructions. The quantity and quality of the recovered DNA samples were determined using a Nanodrop and Qubit^™^ 1× dsDNA HS Assay Kit. The DNA was stored in DNase/Pyrogen-Free water at 4 °C. A pooled PCR-barcoding sequencing library was prepared by an in-house sequencing service (QIB) using the Nextera^™^ DNA Flex Library Preparation Kit (Illumina Inc., Cambridge, UK) with 14 PCR cycles, followed by sequencing using 2 × 150 bp paired-end chemistry (PE150) on an Illumina NovaSeq 6000 platform (Novogene Ltd., Cambridge, UK). The raw data from ME/CFS- and SHHC-derived samples had a Q30 score of >91% per sample and generated 1,252,729,390 paired-end raw sequencing reads (104,394,115.8 ± 49,361,687 per sample, mean ± SD, *n* = 12). Paired-end sequencing reads were provided in FASTQ format. All raw sequencing reads were pre-processed as described above. No positive and negative control samples were included in WMS analysis.

### 4.7. Analysis of VLP and WMS HQ-vOTUs

#### 4.7.1. Genome Assembly and Viral Mining

Cleaned reads were assembled using MEGAHIT assembler (v1.2.9) [80] with default parameters. QUAST (v5.0.2) [81] and SeqFu (v1.8.4) [82] were used to assess the quality and quantity of assembled genomic contigs. Putative uncultivated virus genomes (UViGs) were predicted using VirSorter 2 (v2.0) [83] and VirFinder (v1.1) [84]. All putative viruses and proviruses were sorted and classified into VirSorter 2, and, in parallel, all contigs were run through VirFinder with those having a score of >0.7 and *p* < 0.05 considered viral [6]. All UViGs were pooled and CheckV (v0.7.0) [85] was used to evaluate the quality of both VLP- and WMS-derived UViGs in ‘end-to-end’ mode (with default parameters).

#### 4.7.2. Dereplicating and Generating Non-Redundant Viral Operational Taxonomic Units (vOTUs)

A rapid genome clustering script computing the pairwise average nucleotide identities (ANI) was used to cluster and dereplicate the UViGs, followed by the generation of non-redundant viral operational taxonomic units (vOTUs) clustered at 95% ANI over 85% of the length of sequences, as described by Nayfach et al. (2021) [85] (code available at https://bitbucket.org/berkeleylab/checkv/src/master/, accessed on 13 December 2022). Each vOTU was deemed a single viral species and was represented in the datasets by the longest sequence. vOTUs < 1 Kbp, derived from both the VLP and WMS datasets, were removed. Using CheckV to assess the completeness of vOTUs, low-quality (i.e., <50% genome completeness) and undetermined vOTUs were removed, and the remaining vOTUs, including complete genomes, high-quality (i.e., >90% completeness) and medium-quality genomes (i.e., 50–90% completeness), were retained. Finally, high-quality, non-redundant vOTUs (HQ-vOTUs) were identified. These HQ-vOTUs met certain quality thresholds (i.e., 50–100% genome completeness) and were more likely ‘true’ viruses and were used for read mapping, cluster analysis, taxonomic annotation, and virus–host prediction.

#### 4.7.3. Similarity Identification between VLP and WMS HQ-vOTU Datasets

To determine the sequence similarity between the VLP and WMS HQ-vOTU datasets, BLAST+ (v2.12.0) [86] was used to align the WMS-HQ-vOTUs with the VLP-HQ-vOTUs. Query sequences with the best alignment hit were obtained, followed by the use of filtering thresholds including e-value (cut-off: 1 × 10^−5^), bitscore (cut-off: 1000), percentage of identical matches (pident; 95% cut-off) and query coverage per subject sequence (qcovs; 85% cut-off) to retrieve alignments that were highly likely to be identical between the VLP and WMS datasets.

#### 4.7.4. Detecting Provirus and Plasmid from the VLP and WMS vOTU Datasets 

geNomad (v1.3.3) [87,88] was used to detect potential proviruses (prophages) and plasmids in both the VLP and WMS HQ-vOTU datasets. geNomad, alongside its database (v1.2), was run in ‘end-to-end’ mode with a --min-score of 0.7 and a --cleanup flag.

#### 4.7.5. Read Mapping, Average Coverage Calculation and Count Table Construction

Cleaned reads from 17 VLP-derived samples and 12 WMS-derived samples were separately mapped to the VLP and WMS HQ-vOTUs, respectively, using BWA (v0.7.17) [89] in the ‘bwa-mem’ mode in a paired-end manner, followed by the use of SAMtools (v1.12) [90] to sort and index the resulting alignments. To determine the ‘true’ distribution of each virus (i.e., a virus uniquely originating from either ME/CFS- or SHHC-derived samples or commonly present in both samples), the average read coverage (i.e., [read count × average read length]/vOTU length) was calculated using the ‘average-coverage’ module of BamToCov (v2.7.0) [91] with default parameters. A cut-off threshold of average coverage was set at 10-fold; that is, viruses (vOTUs) were considered ‘truly’ present in a sample when the average coverage was over 10-fold (Appendix A). This analysis was also applied to the vOTU cluster analysis and host predictions. In parallel, both VLP and WMS HQ-vOTUs were merged and deduplicated, followed by read mapping for each sample and generation of a jointed average coverage-based count table using BamToCov for viral abundance and diversity analyses.

#### 4.7.6. vOTU Clustering and Taxonomic Annotation

Cluster analysis to taxonomically classify VLP-HQ-vOTUs and WMS-HQ-vOTUs was performed using vConTACT 2 (v0.9.19) [55,92]. vConTACT 2 used vOTU genomes and reference viral genomes as nodes, with their edges weighted based on the amino acid homology between the two genomes (nodes). HQ-vOTUs and bacteriophage isolate reference genomes from the INPHARED-based reference viral genomes [93] were then clustered to form networks based on their shared gene content and sequence similarity [55,92].

Pharokka (v1.1.0) [94] was used to predict phage genes within the vOTU genomes in ‘meta’ mode and to generate amino acid files in .gbk format; then, both a curated ‘amino acid’ (.faa) file and a ‘gene-to-genome’ (.csv) mapping file were generated using the Perl script (code available at https://github.com/RyanCook94/Random-Perl-Scripts [95], accessed on 13 December 2022). The curated files were then concatenated with INPHARED-based files [93] (ver. 1Dec2022; available at https://github.com/RyanCook94/inphared [96], accessed on 13 December 2022). The merged ‘amino acid’ and ‘gene-to-genome’ mapping files were used as inputs for vConTACT 2 with the Diamond, MCL and ClusterONE modules (with --db ‘none’). The output (c1.ntw) was then visualised using Cytoscape (v3.9.1) [97]. Some original VCs were divided into subclusters by vConTACT 2, where clusters were deemed to be ICTV (sub)families, and subclusters were deemed to be ICTV genera. The clustering status for every vOTU and the reference viral genome was also labelled by vConTACT 2, as described previously [92,98].

To automatically perform taxonomic annotation when using the INPHARED reference viral database [96], graphanalyzer (v1.5.1) [98] was used for the taxonomic assignments of both VLP and WMS HQ-vOTUs; it determines an appropriate taxonomy for each vOTU relying on the ‘weight score’ and ‘clustering status’ generated by vConTACT 2. The ‘status’ clustered, ‘level’ Cx and ‘high weight’ are considered as higher reliability, and ‘level’ Nx and ‘low weight’ have less confidence [99]. Taxonomy is inherited until the ‘subfamily’ level if the status is labelled as ‘Clustered/Singleton’ or ‘Overlap’. It is inherited at the ‘family’ level if vOTUs and reference genomes are not found in the same cluster (i.e., lower levels were therefore omitted and labelled as ‘O’). ‘Outlier’ is probably at the (sub)family level, but this group is less confident. ‘Singleton’ is unable to be included in a network due to a lack of sufficient information for annotation (i.e., labelled as ‘n.a.’). For the vOTUs labelling ‘Clustered’, their taxonomies were inherited until the genus level with high confidence.

#### 4.7.7. Detection of Human Eukaryotic DNA Viruses

Briefly, paired-end viral reads mapped to the VLP and WMS HQ-vOTUs were retrieved and sorted using SAMtools, and then BAM was converted to the FASTQ format using BEDtools (v2.31.0) [100]. Retrieved mapped viral reads were classified using Kraken 2 against the latest RefSeq viral database (available at https://benlangmead.github.io/aws-indexes/k2, accessed on 13 December 2022). In parallel, the presence of eukaryotic viruses was determined in the unknown or unclassified groups of both the VLP and WMS HQ-vOTU datasets using VIRify [101] (with default parameters).

### 4.8. Analysis of Bacteriome

Briefly, cleaned reads were pooled and co-assembled across 12 WMS-derived samples using MEGAHIT (v1.2.9). A total of 774,887 contigs were obtained with an N50 length of 890 bp (Appendix A). Contigs were dereplicated and taxonomically classified using CAT (Contig Annotation Tool) [102]. Reads were then mapped against pooled non-redundant contigs using BWA, sorted and aligned using SAMtools. A count table was generated using BamToCov. Read counts were normalised to genome length per contig per sample for bacterial abundance and diversity analyses.

### 4.9. Host Predictions for VLP and WMS HQ-vOTUs

To predict bacterial hosts for both the VLP and WMS HQ-vOTU datasets, iPHoP (v1.2.0) [103] was used to produce the output of the host predictions. First, total contigs from whole metagenomes were binned into metagenome-assembled genomes (MAGs) separately for each sample using metaWRAP (v1.3.2) [104] which incorporates three binning tools: MetaBAT2 (v2.12.1) [105], MaxBin2 (v2.2.6) [106] and CONCOCT (v1.1.0) [107]. CheckM (v1.2.0) [108] was integrated into the workflow of the metaWRAP pipeline to assess genome completeness and contamination in the ‘bin_refinement’ module. The final binned MAGs were retained based on the criteria of ≥90% completeness and ≤5% contamination using CheckM as previously described [109]. A total of 319 non-redundant binned prokaryotic MAGs (including 317 bacterial and two archaeal genomes) were obtained, with 8–40 MAGs generated from the WMS-derived samples (26.6 ± 12 MAGs per sample, mean ± SD, *n* = 12).

Next, taxonomic decorate tree files derived from MAGs were generated using the ‘de_novo_wf’ module of GTDB-Tk along with its database (ver. release 207) [110], followed by the creation of a customised host database by adding MAGs with the default database using the ‘add_to_db’ module of iPHoP with default parameters. Finally, iPHoP was used to predict the bacterial hosts for both VLP and WMS HQ-vOTUs against customised host databases, with a confidence score of ≥90 (medium or high confidence) being used as a cut-off for the prediction [103]. All predicted bacterial taxa were derived from the GTDB-Tk database, and those genera with a suffix name could be considered as different categories. Virus–host interactions were then investigated to determine potential disease-associated hosts that interacted with ME/CFS-unique viruses present in ME/CFS-derived samples, based on the determination of average read coverage, as described above.

### 4.10. Statistical Analysis

Several statistical techniques were used to compare the viromes and bacteriomes of severely affected ME/CFS patients, SHHC subjects, and processed control samples. All statistical analyses were performed in R (v4.2.0) using phyloseq (v1.40.0), microbiome (v1.18.0), and vegan (v2.6-4) packages. Beta diversities were calculated using Bray–Curtis distance to compare abundances at the genus level, and Jaccard distance was used to count the number of species in common. Both distance measures were visualised using non-metric multidimensional scaling (NMDS) or principal coordinate analysis (PCoA), with the first and second axes shown.

The statistical significance of the similarities in beta diversity within groups was tested using PERMANOVA with appropriate permutation restrictions. Alpha diversity was measured in each dataset using the observed average read coverage of species, Chao1 richness and Shannon index. Data was either rarefied to the minimum sample depth (to compare observed species numbers) or unrarefied (for Shannon index and relative abundance calculations) as appropriate, depending on the specific research question. The specific methods are detailed in the main text, along with each analysis. Linear mixed models were used to compare univariate characteristics (i.e., diversity measures and individual taxonomic abundances) between samples, considering household pairing and individual participant as random effects, as appropriate. Centred log-ratio (CLR) transformations were used prior to statistical comparisons of the taxon abundances. The statistical significance was tested using a two-tailed *t* test for the comparisons of VLP counts and DNA yields between the VLP and WMS groups.

## 5. Conclusions

Using different isolation methodologies and VLP-enriched and whole metagenome-based sequencing methods to obtain more diverse and high-quality vOTUs with high read coverage and high genome completeness, has provided a more comprehensive profile of the human GI DNA virome in healthy individuals and severely affected ME/CFS patients. Although no significant differences were observed in viral or bacterial abundance and/or diversity between in the small cohort of ME/CFS and SHHC, differential effects were evident in microbes from VLP-enriched samples compared to WMS-derived samples. In addition, we were able to predict a possible ME/CFS-associated bacterial host able to interact with ME/CFS-unique viruses observed in patient samples. Therefore, this study provides a framework and rationale for studies in larger cohorts of patients with ME/CFS to further investigate ME/CFS-associated interactions between the intestinal virome and the bacteriome.

## Figures and Tables

**Figure 1 ijms-24-17267-f001:**
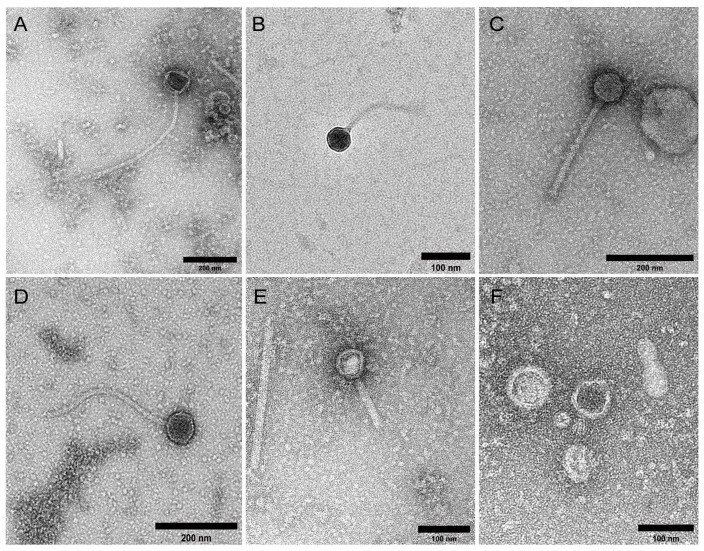
Transmission electron micrographs of faecal VLPs. (**A**–**F**) Representative transmission electron micrographs of VLPs in faecal filtrates derived from one patient and three SHHC subjects. In all samples, the majority of intact virions and VLPs detected belonged to the class *Caudoviricetes* whose morphologically includes siphoviruses (long tails), myoviruses (long, contractile tails) and podoviruses (short tails); a filamentous virus was seen in one sample (**E**). Scale bar = 100–200 nm.

**Figure 2 ijms-24-17267-f002:**
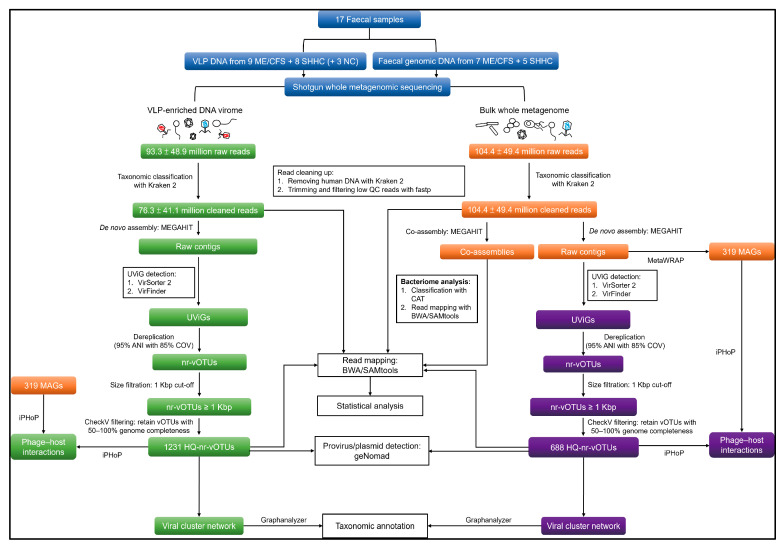
Overview of bioinformatic pipeline for analysing DNA viromes derived from VLP-enriched metagenomes; and WMS-vOTUs and WMS-Bac derived from whole metagenomes. Blue boxes represent the inputs to the pipeline including enriched isolated VLP DNA and faecal genomic DNA derived from the same samples. Three negative controls were included in the enriched VLP analysis. Green boxes represent the process used to analyse the enriched DNA virome. Orange boxes represent the process used to analyse bacteriomes (WMS-Bac), including prokaryotic metagenome-assembled genome (MAG) binning for host prediction. Purple boxes denote the process used to analyse non-enriched viral genomes derived from whole metagenomes (WMS-vOTUs). The solid lines denote primary analyses connecting with boxes representing further analyses.

**Figure 3 ijms-24-17267-f003:**
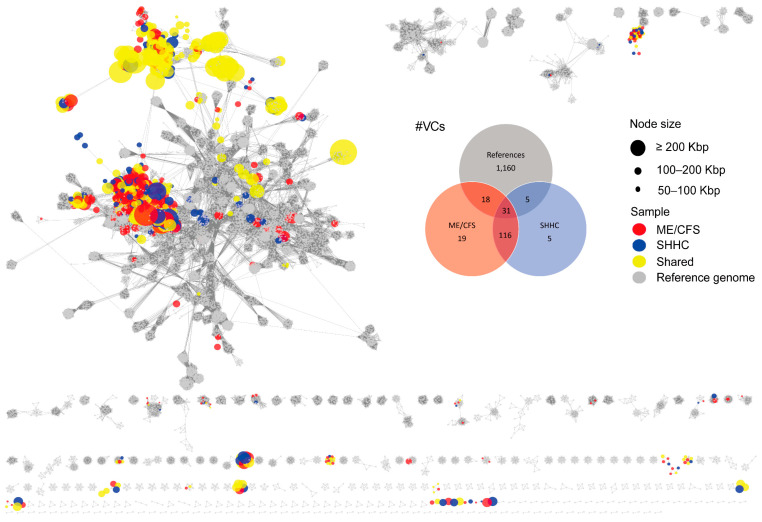
Viral cluster networks derived from the VLP dataset showing similarity between high-quality vOTUs and reference viruses. The networks denote subsets of HQ-vOTUs from the VLP-enriched datasets. HQ-vOTUs were grouped into a viral cluster or subcluster based on genome similarity and amino acid homology against INPHARED-based reference viruses (grey). Based on the mean coverage of mapped reads per vOTU per dataset, each virus (vOTU) was determined based on it being present in at least one dataset. The coloured circles (nodes) denote the true presence of viruses (red: uniquely present in ME/CFS-derived samples; blue: uniquely present in SHHC-derived samples; yellow: communal viruses seen in both samples). The edges (grey lines) represent similarities weighted by edge betweenness across the genomes, with the node size reflecting genome length in Kbp. Venn diagrams display the distribution of VCs in samples, based on the mean coverage of mapped reads (red: VCs composed of ME/CFS-unique viruses; blue: VCs composed of SHHC-unique viruses; grey: VCs composed of reference viral genomes).

**Figure 4 ijms-24-17267-f004:**
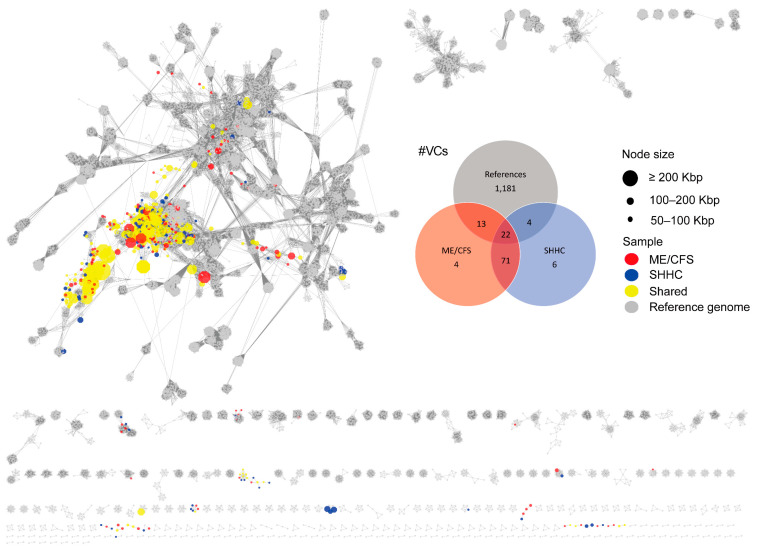
Viral cluster networks derived from the WMS dataset showing similarity between high-quality vOTUs and reference viruses. The networks denote subsets of HQ-vOTUs from the WMS-derived datasets. HQ-vOTUs were grouped into a viral cluster or subcluster based on genome similarity and amino acid homology against INPHARED-based reference viruses (grey). Based on the mean coverage of mapped reads per vOTU per dataset, each virus (vOTU) was determined based on it being present in at least one dataset. The coloured circles (nodes) denote the true presence of viruses (red: uniquely present in ME/CFS-derived samples; blue: uniquely present in SHHC-derived samples; yellow: communal viruses seen in both samples). The edges (grey lines) represent similarities weighted by edge betweenness across the genomes, with the node size reflecting genome length in Kbp. Venn diagrams display the distribution of VCs in samples, based on the mean coverage of mapped reads (red: VCs composed of ME/CFS-unique viruses; blue: VCs composed of SHHC-unique viruses; grey: VCs composed of reference viral genomes).

**Figure 5 ijms-24-17267-f005:**
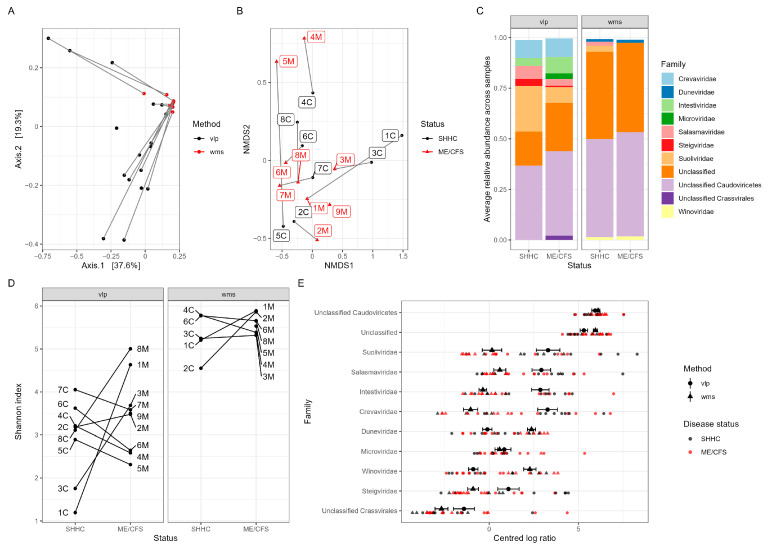
Macrodiversity analyses of intestinal viromes. (**A**) Analysis of beta diversity of intestinal DNA phages from the VLP-enriched (black) and WMS-derived (red) datasets. The VLP and WMS isolation methods generated different viral compositions forming two distinct clusters in the ordination analysis (*p* < 0.001). (**B**) Using NMDS analysis, some household-matched pairs had similar viral compositions while others were more distinct (*p* = 0.172 for overall test of clustering within the household-matched pairs). (**C**) The 11 most abundant intestinal viruses at the family level across the samples separated by the VLP- and WMS-derived groups (unclassified categories included). Each assigned family is depicted by different colours, with the average relative abundance per dataset shown as %. (**D**) Estimation of the alpha diversity of intestinal viromes using the Shannon index within household-matched pairs. Diversity was higher for WMS-derived viruses than VLP-enriched viruses (*p* < 0.0001). (**E**) Analysis of centred log-ratios (CLR) depicting differences in relative abundance estimates of different viral families from VLP-enriched and WMS-derived datasets. The mean CLR of viruses from VLP-enriched datasets are shown as large black circles with the mean CLR of viruses from WMS-derived datasets shown as triangles (error bars representing ± 1 standard error) with individual ME/CFS-derived samples shown in red and with SHHC-derived samples shown in black.

**Figure 6 ijms-24-17267-f006:**
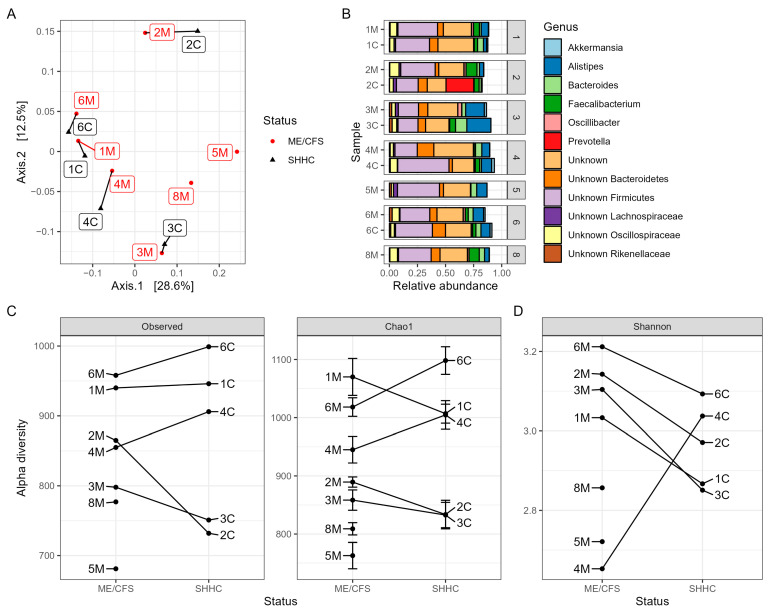
Macrodiversity analysis of intestinal bacteriomes. (**A**) Beta diversity analysis of the bacteriomes from ME/CFS and SHHC groups displayed in a PCoA plot with Jaccard distance showing similarity within household-matched pairs at the genus level (PERMANOVA *p* = 0.002) (M: ME/CFS samples; C: SHHC samples). (**B**) Top 12 relative abundances of bacteria at the genus level (without unknown genera) as derived from five household-matched pairs with two unpaired samples (numbers 1–8 for each plot stand for the pair number of household-matched samples). Each assigned genus is depicted in a different colour. (**C**,**D**) Estimation of the alpha diversity indices. The numbers for average read coverage are observed alongside the estimations of Chao1 richness (**C**) and Shannon index (**D**).

**Figure 7 ijms-24-17267-f007:**
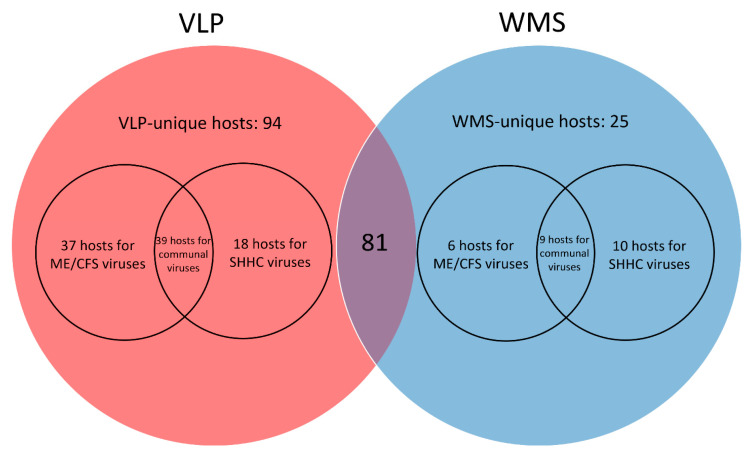
Venn diagrams depicting predicted bacterial hosts from the VLP- and WMS-derived datasets that associate with viruses that are either ME/CFS-unique, SHHC-unique or communal. A total of 81 shared bacterial hosts (middle) were detected in both the VLP- and WMS-derived datasets. Of the VLP-derived bacterial hosts (red), 37 bacterial genera were potentially associated with ME/CFS-unique viruses from the ME/CFS patient samples with six hosts derived from the WMS dataset (blue) associated with ME/CFS-unique viruses.

**Table 1 ijms-24-17267-t001:** Cohort characteristics.

	ME/CFS	SHHC
Age; mean ± SD (median)	33.8 ± 13.0 (37.0)	53.6 ± 15.5 (58.5)
Gender (male:female)	0:9	3:5
Antibiotic use < 4 weeks	0	0
Probiotic capsule supplement use < 4 weeks	0	0
Illness duration > 6 months ^a^	9	N/A ^b^
Self-reported with GI disorder	Four self-reported; Five non-reported	0

^a^ All severely affected patients received their confirmed diagnosis at least six months prior to sample collection. ^b^ N/A: not applicable.

**Table 2 ijms-24-17267-t002:** Number of proviral genomes and plasmid sequences between pre- and post-filtration.

	Proviruses		Plasmids	
	All VLP-nr-vOTUs (*n* = 192,146)	All WMS-nr-vOTUs (*n* = 184,317)	All VLP-nr-vOTUs (*n* = 192,146)	All WMS-nr-vOTUs (*n* = 184,317)
Pre-CheckV filtration	254	537	4119	5359
	VLP-HQ-vOTUs (*n* = 1231)	WMS-HQ-vOTUs (*n* = 688)	VLP-HQ-vOTUs (*n* = 1231)	WMS-HQ-vOTUs (*n* = 688)
Post-CheckV filtration	123	254	6	6

nr: non-redundant; HQ: high quality.

**Table 3 ijms-24-17267-t003:** The 20 most abundant ME/CFS-derived bacterial genera associating with unique or communal viruses in VLP and WMS samples.

Top 20 Bacterial Genera	Hosts Shared in Both VLP and WMS		Hosts in VLP Only		Hosts in WMS Only
*Alistipes*	√				
*Faecalibacterium*	√				
*Bacteroides*	√				√ ^d^
*Oscillibacter * ^a^	−		−		−
*Bifidobacterium*	√				
*Ruminococcus * ^b^	√		√	√	
*Clostridium*	√		√		
*Parabacteroides*	√				
*Akkermansia*	√				
*Collinsella*	√				
*Dialister*	√				
*Roseburia*	√				
*Odoribacter*			√		
*Coprococcus*			√		
*Sutterella*	√				
*Eggerthella*	√				
*Anaerotruncus*			√		
*Adlercreutzia*	√				
*Eubacterium * ^c^	√	√	√	√	

^a^ The genus *Oscillibacter* was not found in any of the prediction outputs. ^b^ Two different *Ruminococcus* are present in the VLP dataset. ^c^ Two *Eubacterium* were shared in both datasets, with another three only seen in the VLP dataset. ^d^ Bacterial hosts associating with ME/CFS-unique viruses are shown in green, with one *Eubacterium* associating with SHHC-unique viruses shown in blue. A checkmark (√) stands for the top 20 bacterial hosts present in the dataset; a minus sign (−) represents the top 20 bacterial hosts are not present in the dataset.

## Data Availability

The sequences derived from the VLP-enriched metagenomes have been deposited in ENA (https://www.ebi.ac.uk/ena/browser/home, accessed on 31 July 2023) under accession no. PRJEB57952 and sequences from the whole metagenomes (WMS) have also been deposited in ENA under accession no. PRJEB57953.

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
