# Peer review of "Investigating the Human Intestinal DNA Virome and Predicting Disease-Associated Virus–Host Interactions in Severe Myalgic Encephalomyelitis/Chronic Fatigue Syndrome (ME/CFS)"

_ijms, 2023, doi:10.3390/ijms242417267_

Round 1
Reviewer 1 Report
Comments and Suggestions for Authors
The study evaluated the virome of individuals with myalgic encephalomyelitis/chronic fatigue syndrome along with the healthy household members. The study is unique in the cohort studied, but also that the authors considered analyzing both VLP and shotgun metagenomics data, which indeed provide a more comprehensive view of extracellular and intracellular viruses, which may counteract the relatively small n size, which is something that the authors addressed when they calculated the number of household needed in the future. My major comments are around how the figures are presented. Overall, while the analyses are robust, the figures NEED to be modified for easier interpretation. See below:
Comments on the Quality of English Language Table 2: Modify header to "Number of Proviral genomes and plasmid sequences pre- and post-filtration. The symbols in Table 3 are not intuitive and are confusing. Since the legend information is also in each of the headers, the authors can replace the blacks dots with a checkmark or something similar that wouldn't require a legend. The same for the empty dots. The idea would be to make the table more intuitive and easy to read. As of now, this table is hard to read and interpret. Figure 3: The networks are blurry and impossible to interpret. The authors need to find the way to make these data interpretable. Figure 4C: The percentages may not be necessary. Figure 4E: Can the authors color the CLR based on disease state and shape by method instead? Minor comments: Lines 176-178: Can the authors speculate why this is the case? Can they mention here if the same amount of stool sample and protocol was used for the extractions? Line 323: "were potentially"English is ok
Author Response
We would like to thank the reviewers for their considered and thoughtful review of our manuscript. We have below provided a point-by-point response to all comments made by the reviewers and highlighted changes in the manuscript by yellow highlighting. We also include below the relevant line numbers where revisions have been made.
Reviewer 1
Major comments:
- Table 2: Modify header to "Number of Proviral genomes and plasmid sequences pre- and post-filtration.
Revised accordingly.
- The symbols in Table 3 are not intuitive and are confusing. Since the legend information is also in each of the headers, the authors can replace the blacks dots with a checkmark or something similar that wouldn't require a legend. The same for the empty dots. The idea would be to make the table more intuitive and easy to read. As of now, this table is hard to read and interpret.
Following the reviewer’ insghtful comment, we have replaced the dots with a checkmark and also highlighted those bacterial hosts associating with ME/CFS-unique viruses in green, with one Eubacterium associating with SHHC-unique viruses shown in blue.
- Figure 3: The networks are blurry and impossible to interpret. The authors need to find the way to make these data interpretable.
We have divided the original Figure 3 (3A and 3B) into Figure 3 and Figure 4, to enable the figures to be enlarged for improved clarity. We have also highlighted the differences between both networks/datasets in the text, based on the distributions of their vOTU genomes, genome sizes and VCs in samples (Lines 204–206).
- Figure 4C: The percentages may not be necessary. Figure 4E: Can the authors color the CLR based on disease state and shape by method instead?
These modifications have been made to the manuscript. As we now included a new Figure 4, this is nowFigure 5.
Minor comments:
Lines 176-178: Can the authors speculate why this is the case? Can they mention here if the same amount of stool sample and protocol was used for the extractions?
- We speculate that the very low numbers of non-human reads associated with bacterial or viral taxa in the NC samples were likely derived from sequencing and/or amplification biases or, kit/reagent contamination. While these contaminants or biases may be difficult to completely exclude from sequence datasets, we still obtained high quality vOTUs of high purity which were absent in the NC samples (Lines 564–567). By adopting stringent criteria and approaches to analysing sequence data it is possible to minimise or even exclude potential contaminations or biases to obtain high confidence viral taxonomies.
- Three extraction buffer-only (negative) control samples were included as process-controls (with no spiking with stool samples) and processed in parallel with samples for VLP- and DNA-isolations using the same protocol. This information has been added to the revised manuscript (Lines 176–178 and Lines 693–695).
Reviewer 2 Report
Comments and Suggestions for Authors
The reviewer does appreciate the authors contributed a fantastic manuscript, which combined the DNA virome from isolated virus-like particle (VLP) and the whole metagenome from the same stool sample from a small group of healthy individuals and patients with severe myalgic encephalomyelitis/chronic fatigue syndrome (ME/CFS) to identify the core viral group while predicting interactions between anaerobic members and the unique viruses present in the ME/CFS microbiome. The result present and the manuscript wrote generally well, while only having minor improvements.
Figure 1 should be marked A, B, C…
Please indicated the letter Y mean in Supplementary S1?
Author Response
We would like to thank the reviewer for their considered and thoughtful review of our manuscript. We have below provided a point-by-point response to all comments made by the reviewer and highlighted changes in the manuscript by yellow highlighting. We also include below the relevant line numbers where revisions have been made.
Minor comments:
- Figure 1 should be marked A, B, C…
This information has been added to the manuscript.
- Please indicated the letter Y mean in Supplementary S1?
This information has been added to the Supplementary Table S1 (spreadsheet 1).
Round 2
Reviewer 1 Report
Comments and Suggestions for Authors
Authors seemed to have addressed the comments.
Comments on the Quality of English LanguageQuality seems good.